# Immunomodulatory Aspects of Therapeutic Plasma Exchange in Neurological Disorders—A Pilot Study

**DOI:** 10.3390/ijms24076552

**Published:** 2023-03-31

**Authors:** Fabian Foettinger, Georg Pilz, Peter Wipfler, Andrea Harrer, Jan Marco Kern, Eugen Trinka, Tobias Moser

**Affiliations:** 1Department of Neurology, Christian Doppler University Hospital, European Reference Network EpiCARE, Paracelsus Medical University, 5020 Salzburg, Austria; 2Department of Dermatology and Allergology, Paracelsus Medical University, 5020 Salzburg, Austria; 3Department of Clinical Microbiology and Hygiene, Paracelsus Medical University, 5020 Salzburg, Austria; 4Center for Cognitive Neuroscience, Neuroscience Institute, Christian Doppler University Hospital, Paracelsus Medical University, 5020 Salzburg, Austria

**Keywords:** plasma separation, mode of action, antibody titers, interleukin, multiple sclerosis, cytokines

## Abstract

Therapeutic plasma exchange (TPE) is used for drug-resistant neuroimmunological disorders, but its mechanism of action remains poorly understood. We therefore prospectively explored changes in soluble, humoral, and cellular immune components associated with TPE. We included ten patients with neurological autoimmune disorders that underwent TPE and assessed a panel of clinically relevant pathogen-specific antibodies, total serum immunoglobulin (Ig) levels, interleukin-6 (IL-6, pg/mL), C-reactive protein (CRP, mg/dL), procalcitonin (PCT, µg/L) and major lymphocyte subpopulations (cells/µL). Blood was collected prior to TPE (pre-TPE, baseline), immediately after TPE (post-TPE), as well as five weeks (follow-up1) and 130 days (follow-up2) following TPE. Pathogen-specific antibody levels were reduced by −86% (*p* < 0.05) post-TPE and recovered to 55% (follow-up1) and 101% (follow-up2). Ig subclasses were reduced by −70–89% (*p* < 0.0001) post-TPE with subsequent complete (IgM/IgA) and incomplete (IgG) recovery throughout the follow-ups. Mean IL-6 and CRP concentrations increased by a factor of 3–4 at post-TPE (*p* > 0.05) while PCT remained unaffected. We found no alterations in B- and T-cell populations. No adverse events related to TPE occurred. TPE induced a profound but transient reduction in circulating antibodies, while the investigated soluble immune components were not washed out. Future studies should explore the effects of TPE on particular cytokines and assess inflammatory lymphocyte lineages to illuminate the mode of action of TPE beyond autoantibody removal.

## 1. Introduction

Autoimmune disorders can affect any part of the central and the peripheral nervous system (CNS and PNS). Moreover, they differentially involve the humoral and cellular components of the immune system [1,2]. Irrespective of the underlying pathophysiology, therapeutic plasma exchange (TPE) represents a highly effective rescue treatment for patients with steroid-refractory and severe immunological conditions [3]. TPE is an extracorporeal procedure that involves the removal, separation and replacement of human plasma and represents, by its nature, a nonspecific treatment approach. In fact, the clinical benefit of TPE has been demonstrated for various neurological diseases such as Guillain–Barre syndrome (GBS) and chronic inflammatory demyelinating polyneuropathy (CIDP), myasthenia gravis (MG), neuromyelitis optica spectrum disorders (NMOSD) and autoimmune encephalitis (AE) [4,5,6]. Nevertheless, knowledge about the immunomodulatory aspects of TPE remains sparse. The most intuitive rational behind this procedure is the separation of pathogenic substances from the blood and especially that of removing autoreactive antibodies. However, how symptom relief is achieved in disorders that are primarily considered to be lymphocyte mediated such as multiple sclerosis (MS) and in which autoantibodies appear to play a minor role remains poorly defined [7]. Moreover, conflicting evidence, mostly deriving from animal studies, suggested an overproduction of antibodies following plasma separation, which in turn would bear the potential to induce to a clinical rebound [8,9,10]. It is largely unknown how long TPE effects last, and there is no consensus on when subsequent immunomodulatory maintenance treatments should be initiated.

The aim of this pilot study was to evaluate the extent and duration of the immunomodulatory effects induced by TPE. For this reason, we longitudinally explored the TPE-associated serologic dynamics of pathogen-specific antibody-levels and immunoglobulin classes, of lymphocyte counts and that of soluble components of the immune system including interleukin-6 (IL-6) among a cohort of 10 patients diagnosed with neurological autoimmune disorders.

## 2. Results

### 2.1. Study Population

Of the 10 patients enrolled, three were woman. The median age of the cohort was 53 years (interquartile range (IQR) 36–72 years). Patient characteristics and indications for TPE are listed in Table 1. Eight patients received five cycles of TPE, one patient with acute disseminated encephalomyelitis (ADEM) received four and one patient with steroid-refractory ON received three cycles. TPE was well tolerated in all patients and no adverse effects were observed. Patients were followed up at a median of 37 days (IQR 31–43 days, follow-up1) and at 130 days (range 76–185 days; follow-up2) from the last TPE session.

### 2.2. Washout of Pathogen-Specific Antibodies

The average washout of pathogen-specific antibodies from pre-TPE baseline values to post-TPE was −86% (95% CI: 83–89%, *p* < 0.0001). This effect was transient and antibody levels recovered to an average of 55% (95% CI: 45–66%, *p* = 0.0019) at follow-up1 and reached pre-TPE baseline values at follow-up2. Pathogen-specific antibody dynamics associated with TPE are graphically illustrated in Figure 1. The highest decline of antibody levels was observed in varicella zoster virus (VZV)-IgG, with an average removal of −92% (95% CI: 88–96%, *p* = 0.0084). The highest recovery rate was found in tick-borne encephalitis (TBE)-IgM, as antibody levels were reduced by −88% (95% CI: 80–96%, *p* = 0.071) at post-TPE and approached baseline values at follow-up1 (81%, 95% CI: 34–128%, *p* = 0.9422). Slowest recovery rates were observed in hep-B antibody levels, as the follow-up1 concentrations only increased to an average of 27% (95% CI: −21–75%) compared to pre-TPE baseline levels.

### 2.3. Dynamics of Immunoglobulins

Dynamics of immunoglobulin washout behaved similar to those of pathogen-specific antibodies as shown in Figure 2, while the recovery kinetics were found to be more pronounced. The average removal of immunoglobulin referenced by the respective baseline values from pre-TPE to post-TPE was −85% (95% CI: 82–88%, *p* < 0.0001). At follow-up1, the measured antibody levels recovered to an average of 80% (95% CI: 68–91%, *p* = 0.1372) compared to pre-TPE levels. Data at follow-up2 suggest that antibody levels fully recover to pre-TPE baseline state after few months (average Ig concentration referenced by baseline values 97%; 95% CI: 72–121%, *p* > 0.05). IgG was the most profoundly affected IgG, while the impact of TPE on IgA and IgM was less pronounced. These latter subclasses almost completely recovered at follow up-1 (Figure 2a–c).

The mean pre-TPE to post-TPE changes amongst IgG1-4 subtypes were similar (average reduction −89%, 95% CI: 88–91%, *p* = 0.0006), but we observed differences in the respective recovery dynamics. At follow-up1, IgG3 exhibited the highest recovery rates (recovery to 83%, 95% CI: 38–128% compared to baseline) and IgG2 the lowest (recovery to 47%, 95% CI 21–73; Figure 2d). At follow-up 2, mean IgG4 levels were fully repleted whilst IgG1-3 concentrations were still below pre-TPE values.

### 2.4. Impact of TPE on IL-6 and CRP and Main Lymphocytes

Serum concentrations of IL-6 increased in 7 out of 9 patients, and the mean IL-6 values augmented from 4.0 pg/mL (95% CI: 0.48–7.72 pg/mL) at pre-TPE to 16.2 pg/mL (95% CI: −3.0–35.2 pg/mL, *p* = 0.1799, Figure 2e) at post-TPE. There were not enough data available throughout the follow-up assessments to allow considerations regarding mid- to long-term effects of TPE on IL-6 levels. Concentrations of CRP increased in four patients and the mean CRP values augmented from 0.4 mg/dL (95% CI: 0.0–0.7 mg/dL) at pre-TPE to 1.3 mg/dL (95% CI: 0.1–2.5 mg/dL, *p* = 0.1184) at post-TPE (Figure 2f). The CRP increase was, however, transient in nature and CRP levels reached normal ranges by follow-up1. In contrast to these results, PCT remained within the normal range in all but one patient (Figure 2g).

We found no significant short- or long-term effects of TPE on absolute counts of B- and T-lymphocytes. Amongst CD3+ T cell subpopulations, TPE was also not associated with changes in absolute numbers of CD4+, CD8+ or HLADR+ T cells (*p* > 0.05, Figure 3).

## 3. Discussion

In the present study, we corroborated the presumed rationale behind TPE [11], i.e., that this procedure removes the bulk of circulating antibodies. After five TPE cycles, the immunoglobulin levels among our cohort were reduced by more than 85% compared to baseline. Five weeks later, antibody levels recovered to approximately 50% of pre-TPE concentrations and reached baseline values at 130 days post-TPE. Even though we analyzed the impact of TPE on total immunoglobulins and on protective antibodies, comparable recovery kinetics of pathogenic autoantibodies have been reported in MG patients [12]. The dynamics of antibody repletion following separation yield clinical implications for the initiation of maintenance immunosuppressive therapy following TPE. Importantly, we did not observe a recovery pattern indicating a TPE-associated antibody overshoot, as suggested by some animal models and in vitro studies to occur within days to a few weeks [8]. An immunoglobulin related rebound phenomenon following TPE was thought to be induced by the abolishment of negative feedback regulations that antibodies exert on plasma cells [3], and would, according to our present understanding of autoantibody-mediated disorders, increase the risk of clinical deterioration. In contrast to the reported antibody dynamics in rabbits, which already reversed 48 h after removal by TPE [9], antibody recovery in humans appears to span several weeks. Although the acute benefits of TPE in inflammatory disorders are likely attributed to the profound elimination of antibodies as demonstrated in this study, the given recovery dynamics in turn imply that it does not qualify as an ideal treatment candidate for long-term remission.

When exploring the impact on different immunoglobulin subsets we found that IgG, the primary substrate of many neurologically relevant autoantibodies [1,2], was most extensively affected. Moreover, levels of IgG1-3 remained depleted throughout the whole study period and did not reach baseline values at follow-up2. Whether the more prominent recovery of IgM and IgA, which are both involved in early host defense [13], contributes to the low infection rates associated with TPE remains to be further explored. However, our data clearly show that despite the transient washout, pathogen-specific antibodies fully recover to baseline values within a few months from TPE. These findings imply that booster vaccinations may not be required in patients undergoing TPE.

The kinetics of circulating antibody levels are regulated by the distribution and production rate of immunoglobulins as well as by their half-lives [14]. Antibodies are synthesized and secreted by terminally differentiated B cells, mostly long-lived plasma cells settled in niches within the bone marrow and secondary lymphoid tissues [13]. As a result, plasma cells and their capacity to produce pathogenic autoantibodies likely remain unaffected by TPE, indicating a short-lasting duration of the clinical TPE benefits. To date, treatment advice considering immunomodulatory approaches and algorithms following TPE is sparse. Our data imply the necessity to take precautions for subsequent maintenance immunotherapy and suggest that these should be considered early on to prevent disease reactivation after TPE. This also applies to NMOSD, an autoimmune disease characterized by severe relapses resulting in permanent disability [15]. Most patients with NMOSD are characterized by seropositivity for antibodies against the astrocyte water channel aquaporin-4 (AQP4), which are considered causative of the disease [15]. These directly pathogenic agents are transiently eliminated by TPE but are continuously reproduced by plasma cells. In fact, B-cell depletion by monoclonal anti-CD20 antibodies is recommended to be administered without a window after TPE in NMOSD patients [16]. Paradoxically, anti-CD20 therapies were proven very effective in NMOSD starting from the first infusion, even though they apparently do not affect plasma cells nor circulating AQP-4 antibodies [17]. The exact mechanism of action that underlies both anti-CD20 therapy and TPE in NMOSD, whereby the former leads to lasting depletion of B cells while having no immediate impact on pre-existing immunoglobulins and humoral immunity [18,19,20], and the latter effectively clears antibodies but spares lymphocyte counts, remains uncertain. This apparent paradox illustrates the incompleteness in our understanding of autoimmune disorders and also emphasizes the complexity of the human immune system. The rationale behind the pronounced treatment effects of both anti-CD20 medications and of TPE not only in NMOSD patients but among a wide range of inflammatory disorders likely consists of a myriad of not yet fully illuminated interactions between the humoral and cellular immune axes. It appears less likely that the beneficial aspects of both treatment approaches are solely explained by pathophysiological differences underlying the acute or the chronic phase of immunological diseases. Yet, as exemplified by MS, relapses and remission are treated distinctively, indicating that heterogeneous processes prevail at different stages of the disease. In fact, steroids and TPE are used to treat acute clinical deteriorations while disease modifying therapies (DMTs) are administered during remission to prevent further relapses [21,22]. Future studies should investigate the impact of anti-CD20 medications and TPE on cytokines and other inflammatory substances to determine the common denominator responsible for the treatment response of both therapies. Such data could not only expand our knowledge on the interplay between humoral and cellular immune processes but also illuminate the mode of action of TPE in presumed non-antibody-mediated disorders. Autoimmunity in MS for example appears primarily lymphocyte driven [7,23] and a causative pathogenic antibody has not been identified [24]. In this regard, we found no effects of TPE on the main lymphocyte subclasses, and the rationale of this intervention in severe MS relapses remains elusive. Immunomodulatory effects of TPE, beyond antibody removal, were hypothesized to include shifts within T helper cell phenotypes, elimination of immunocomplexes and sensitizing lymphocytes for subsequent immunosuppression [3], which we were not able to explore in our analysis. As this pilot study was not performed for in-depth immunophenotyping of T and B cell subsets, we cannot exclude effects of TPE on particular lymphocyte subgroups or on qualitative properties of immune cells that would explain the benefits of TPE in steroid-refractory MS relapses [6,25,26]. Therefore, future studies should not only focus on various proinflammatory immune cells but also explore potential effects of plasma exchange on adhesion molecules, as they are dysregulated in MS and represent a target of DMTs [27,28].

An unexpected observation in our study was that IL-6 concentrations increased in response to TPE. This again appears paradoxical, as IL-6 constitutes a proinflammatory cytokine which is involved in several neurological autoimmune disorders by promoting survival and functionality of antibody-producing cells, by enhancement of T cell proliferation and by disruption of the blood–brain barrier [15]. IL-6 is elevated in NMOSD patients and blockage of its receptor resulted in profound treatment success [15]. Therefore, these data suggest that IL-6 plays a pathogenic role in NMOSD and as such the overproduction reported among this cohort is quite surprising. However, our findings are in line with a study on 10 patients with systemic vasculitis, which also observed augmented IL-6 concentrations following TPE [29]. Yet, to date, evidence exploring the impact of TPE on various cytokines in neurological autoimmune disorders is limited and in part conflicting and therefore not sufficient to abstract final conclusions. A study comprising a cohort of 19 patients with MG for example found significant higher IL-10 levels associated with usage of double-filtration plasmapheresis, while other interleukins remained unaffected [30]. Our study does not allow for elucidation of the nature of the observed IL-6 augmentation and we subsequently cannot exclude an increase related to the TPE procedure itself, as suspected by other reports [29,30,31]. In a small cohort of patients with thrombotic thrombocytopenic purpura, however, IL-8 and TNF-alpha levels, although removed by each TPE session, rebounded the day after separation [31], which indicates that the mechanism behind the associated cytokine overproduction is independent of the acute TPE intervention. To conclude, it appears very unlikely that the method of action of TPE relies on or includes IL-6 removal to hamper or prevent autoimmunity.

The main limitations of this preliminary work stem from the small sample size and especially from the fact that most patients received subsequent IVIG or immunosuppressive therapies, which have profound effects on humoral and cellular components of the immune system. We therefore excluded the follow-up data of those individuals which led to a reduced number of data available at long-term. As patients that suffer from severe inflammatory disorders and that require TPE commonly need a combination of immunotherapies, long-term assessments to study the immunomodulatory effects of TPE will be difficult to perform also in the future.

Another limitation is that we did not explore the impact of TPE on autoantibodies but primarily on protective immunoglobulins.

According to the findings from this pilot project, larger studies should be conducted to assess memory phenotypes of lymphocyte classes and focus on particular cytokines and other inflammatory soluble substances. Cytokines represent critical mediators of immune cell communication and may also contribute to tissue damage in inflammatory disorders [32]. Due to their biological properties, they likely represent a target of TPE and may explain the rational for treatment response in inflammatory disorders beyond autoantibody removal.

## 4. Materials and Methods

### 4.1. Study Population and Recruitment

In this prospective pilot study, we recruited ten patients that received TPE due to neurologic autoimmune disorders between 2021 and 2022. Patients were included from the Christian Doppler University Hospital (Salzburg, Austria) and TPE was performed at the department’s intensive care unit.

Demographic and clinical data were obtained during hospital courses and at routine follow-up visits. Blood was collected prior to TPE (pre-TPE, baseline) and immediately after TPE (post-TPE). Moreover, blood was drawn approximately one month (follow-up1) and approximately four months (follow-up2) after TPE. Once patients received subsequent IVIG or immunosuppressant treatment, further laboratory data were excluded from this study to prevent a possible impact of these therapies on the here investigated parameters.

This study was approved by the local ethics committee (Landesethikkommission Salzburg 1178/2021) and all patients gave written informed consent.

### 4.2. Therapeutic Plasma Exchange

The procedural indication for TPE was addressed by the referring neurologists. TPE was performed at the intensive care unit according to current guidelines [4,33] and institutional protocols as follows: Local citrate anticoagulation was used to prevent clotting of extracorporeal whole blood. During the procedure, patients were regularly assessed for signs and symptoms of hemodynamic instability, and blood pressure, as well as pulse curve were obtained and regularly examined. Patient’s blood plasma was exchanged with 5% human albumin solutions. Patients received three to five cycles of TPE with an interval of one to three days, depending on the recovery of blood coagulation parameters. In one cycle, 3000 mL plasma was exchanged against 3000 mL human albumin solution. Complications and adverse reactions during and after the procedure were thoroughly assessed by on-duty neurologists.

### 4.3. Sample Collection and Laboratory Analysis

Peripheral venous blood was collected in anticoagulated EDTA whole-blood tubes (Greiner Bio-One, Kremsmünster, Austria) for lymphocyte subtyping and native blood tubes (Becton Dickinson, Franklin Lakes, NJ, USA) for the analysis of humoral factors and proinflammatory soluble molecules. All parameters were considered routine laboratory analyses and performed by the local Department of Laboratory Medicine certified according to the ISO-9001 standard and working according to ISO-15189 standards.

Pathogen-specific IgG antibody concentrations to measles, mumps, rubella, VZV, hepatitis B (hep-B), diphtheria, tetanus toxins, TBE, and severe acute respiratory syndrome coronavirus type 2 (SARS-CoV-2) were quantitatively analyzed by chemiluminescent microparticle immunoassay (CMIA) or enzyme-linked immunosorbent assay (ELISA) test. Details on measurement instruments and assay cut-off values are depicted in Table A1. Antibody levels to the eight respective pathogens were available in six patients throughout baseline and follow-up1 and in two patients at follow-up2. Immunoglobulin concentrations (total IgG, IgG subclasses 1/2/3/4, IgA and IgM) were determined by kinetic nephelometry (Siemens BNII device, Siemens Healthineers, Erlangen, Germany). Pre-TPE and post-TPE immunoglobulin concentrations (IgG, IgG 1-4, IgA, and IgM), as well as follow-up1 levels were measured in six patients and follow-up2 values were determined in two patients. Absolute B (CD19+) and T (CD3+) lymphocyte counts were quantified (Sysmex XN system, Sysmex, Kōbe, Prefecture Hyōgo, Japan) and major T-lymphocyte (CD3+) subpopulations including CD4+T helper cells, CD8+T lymphocytes and HLADR+T helper lymphocytes subtyped by means of flow cytometry as previously described (BD FACSLyric, Becton Dickinson, Franklin Lakes, NJ, USA) [18]. Lymphocyte counts were available in five individuals at pre-TPE, in six at post-TPE and in four and two at the respective follow-up1 and 2.

Lastly, IL-6 and procalcitonin (PCT) were determined using electrochemiluminescence (ECL) technology-based immunoassays and c-reactive protein (CRP) via clinical chemistry assays. Assessment of IL-6, PCT and CRP was available in nine patients. The analyses were performed on COBAS 600 (Hitachi, Roche, Vienna, Austria).

### 4.4. Statistical Analysis

Demographics and patient baseline characteristics were descriptively explored. The range of estimates were determined via the delta method for computation of the 95% confidence interval (CI) of geometric means. We measured percentual changes in pathogen-specific antibody levels and immunoglobulin concentration for each patient after TPE referenced by their respective pre-TPE baseline values. We calculated the average percentual removal of alloantibodies and immunoglobulins. Proportional changes in pathogen-specific antibody levels, immunoglobulin concentration as well as absolute counts of immunocompetent cells and IL-6 concentration were plotted as XY data interconnected with lines. Significance testing was performed using paired t-test and overall significance set at *p* < 0.05.

All statistical analysis and graphical illustrations were performed using GraphPad Prism (Version 9.5.0, GraphPad Software, 225 Franklin Street. Fl. 26, Boston, MA 02110, USA).

## 5. Conclusions

TPE leads to a profound but transient removal of circulating antibodies, and precautions for successive maintenance therapy should be taken without delay to prevent permanent disability. On the other hand, the recovery rates of pathogen-specific antibodies together with preserved lymphocyte counts may explain the good safety profile associated with TPE observed among our cohort. Moreover, we found no washout of soluble components of the immune system associated with TPE, especially of IL-6. Future studies should investigate the effects of TPE on specific pro-inflammatory immune cell subsets and on cytokines beyond IL-6 to understand the mode of action in neuroinflammatory disorders that are presumably not antibody mediated.

## Figures and Tables

**Figure 1 ijms-24-06552-f001:**
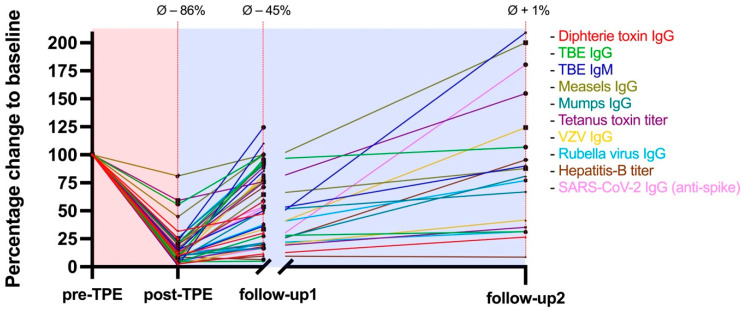
Changes in pathogen-specific antibodies in association with TPE. Antibody levels decreased by −86% compared to baseline values and recovered to 56% percent of pre-TPE baseline values at follow-up1 and approached baseline values at follow-up2. Follow-up1 and follow-up2 were conducted 5 weeks and 130 days after the last TPE session, respectively. Data from individuals treated with immunosuppressive drugs (anti-CD20 monoclonal antibodies *n* = 2; azathioprine *n* = 1) and/or with intravenous immunoglobulins (IVIG; *n* = 5) after TPE were excluded from the respective follow-ups. One patient was lost to follow-up after TPE. TPE = therapeutic plasma exchange; pre-TPE = prior to therapeutic plasma exchange; post-TPE = immediately after therapeutic plasma exchange; Ø = average change to baseline levels; TBE = tick-borne encephalitis; VZV = varicella zoster virus.

**Figure 2 ijms-24-06552-f002:**
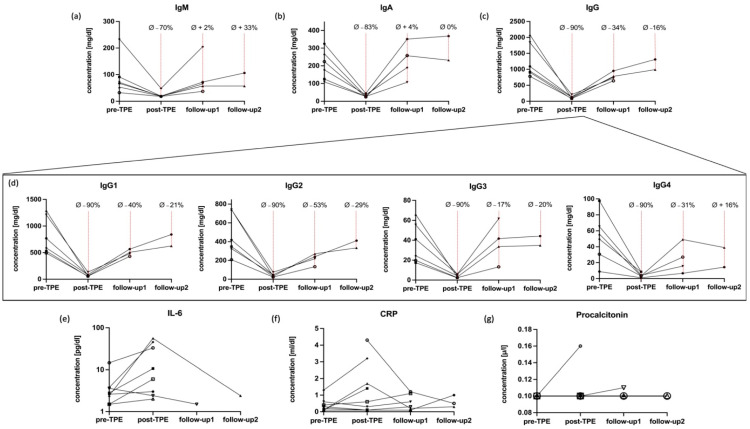
Absolute changes in humoral and soluble components of the immune system associated with TPE. All immunoglobulin subclasses were substantially removed by TPE (**a**–**d**). IgA and IgM, however, recovered faster than IgG. The IgG subclasses 1–3 did not reach baseline values throughout the follow-up, while IgG4 levels fully recovered at follow-up2 (**d**). As for changes in soluble factors, IL-6 and CRP were observed to be increased in response to TPE (**e**,**f**) while procalcitonin was not affected (**g**). Follow-up1 and follow-up2 were conducted 5 weeks and 130 days after the last TPE session, respectively. Data from individuals treated with immunosuppressive drugs (anti-CD20 monoclonal antibodies *n* = 2; azathioprine *n* = 1) and/or with intravenous immunoglobulins (IVIG; *n* = 5) after TPE were excluded from the respective follow-ups (**a**–**e**). One patient was lost to follow-up after TPE. Symbols and connecting lines show the raw data of each patient included at the respective time of sampling. Pre-TPE = prior to therapeutic plasma exchange; post-TPE = immediately after therapeutic plasma exchange; Ø = average change to baseline levels; IL-6 = interleukin 6; CRP = C-reactive protein.

**Figure 3 ijms-24-06552-f003:**
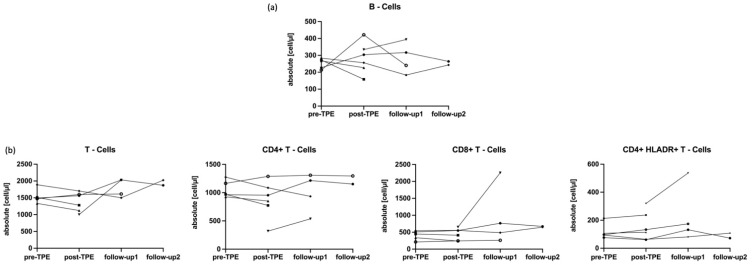
Quantitative changes in major lymphocyte subpopulations associated with TPE. TPE did not impact on B cells (**a**) and had no substantial effects on any of the assessed T lymphocytes (**b**) including T helper cells (CD4+), cytotoxic T cells (CD8+) and proinflammatory T helper cells expressing HLADR. Follow-up1 and follow-up2 were conducted 5 weeks and 130 days after the last TPE session, respectively. Lymphocyte data were available in five individuals at pre-TPE, in six at post-TPE and in four and two at the respective follow-ups. Data from individuals treated with immunosuppressive drugs (anti-CD20 monoclonal antibodies *n* = 2; azathioprine *n* = 1) and/or with intravenous immunoglobulins (IVIG; *n* = 5) after TPE were excluded from the respective follow-ups. One patient was lost to follow-up after TPE. Symbols and connecting lines show the raw data of each patient included at the respective time of sampling. Pre-TPE = prior to therapeutic plasma exchange; post-TPE = immediately after therapeutic plasma exchange.

**Table 1 ijms-24-06552-t001:** Patient characteristics (*n* = 10).

Age in Years, Median (IQR)	53 (36–72)
Female, *n* (%)	3 (30)
Caucasian, *n* (%)	10 (100)
Neurologic disorders:	
Steroid-refractory ON, *n* (%)	3 (30)
CIDP, *n* (%)	2 (20)
GBS, *n* (%)	2 (20)
NMDARE, *n* (%)	1 (10)
MG, *n* (%)	1 (10)
ADEM, *n* (%)	1 (10)
Number of TPE cycles:	
5	8
4	1 *
3	1 **
Average time of symptom onset or relapse to TPE, in days (SD)	22 (±15)
Patients with steroid therapy prior to TPE, *n* (%):	6 (60)
Average time of steroid therapy to TPE, in days (SD)	6 (±5)

IQR = interquartile range; *n* = number of patients; SD = standard deviation; ** ON = optic neuritis; CIDP = chronic inflammatory demyelinating polyneuropathy; GBS = Guillain–Barré Syndrome; MG = myasthenia gravis; NMDARE = NMDA receptor encephalitis; * ADEM = acute disseminated encephalomyelitis.

## Data Availability

The data that support the findings of this study are available from the corresponding author (T.M.), upon reasonable request.

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
