# Peer review of "Immunomodulatory Aspects of Therapeutic Plasma Exchange in Neurological Disorders—A Pilot Study"

_ijms, 2023, doi:10.3390/ijms24076552_

Round 1

Reviewer 1 Report

In this study, the authors explored changes of soluble, humoral, and cellular immune components associated with TPE in ten patients with neurological autoimmune disorders, and pathogen-specific antibody levels were reduced post-TPE and recovered in the follow-ups. Mean IL-6 and CRP concentrations increased while PCT 24 remained unaffected. Furthermore, no alterations in B- and T-cell populations. No adverse events related to TPE occurred. However, a few issues need to be addressed.

1.       In Figure 1, it is better to label the color line, especially VZV-IgG, FSME-IgM and hep-B.

2.       According to Figure 2, the samples were not 10 instead of 6 or smaller even at the pre-TPE point, could the authors explain?

3.       The results provided were not consistent with samples, and the authors should point out the excludes.

4.       The conclusions were not strongly supported by the analysis.

5.       It cannot provide useful information for readers, which made the research not scientific soundness.

Author Response

We would like to really thank the reviewer for the time and suggestions which were very helpful to improve our manuscript.

  1. In Figure 1, it is better to label the color line, especially VZV-IgG, FSME-IgM and hep-B.

This is a very good suggestion, thank you! We have now changed the colours as recommended by the reviewer. Each colour now represents a pathogen-specific antibody as explained in the legend.

  1. According to Figure 2, the samples were not 10 instead of 6 or smaller even at the pre-TPE point, could the authors explain?

You are right, thank you for this point. Unfortunately, lymphocyte subsets were not assessed for all patients at pre-TPE and are available for 5 individuals. We have added the following statement to the respective figure legend:

Lymphocyte data were available in five individuals at pre-TPE, in six at post-TPE and in four and two at the respective follow-ups.

The exact number of patients included at each time of sampling is described in the methods section for the respective assessments.

  1. The results provided were not consistent with samples, and the authors should point out the excludes.

Thank you, this is an important suggestion and we have now added the respective information to each corresponding figure legend, which now reads as follows for figure 1; figure 2a-e and figure 3 :

“Data from individuals treated with immunosuppressive drugs (anti-CD20 monoclonal antibodies n = 2; azathioprine n = 1) and/or with intravenous immunoglobulins (IVIG; n = 5) after TPE were excluded from the respective follow-ups. One patient was lost to follow-up after TPE.”

  1. The conclusions were not strongly supported by the analysis.

We would like to refer to the response of the next point (5.).

  1. It cannot provide useful information for readers, which made the research not scientific soundness.

We have now changed the conclusion, to make it more practical for the readers. The respective paragraph now reads as follows:

TPE leads to a profound but transient removal of circulating antibodies, and pre-cautions for successive maintenance therapy should be taken without delay to prevent permanent disability. On the other hand, the recovery rates of pathogen-specific anti-bodies together with preserved lymphocyte counts may explain the good safety profile associated with TPE observed among our cohort. Moreover, we found no wash-out of soluble components of the immune system associated with TPE, especially of IL-6. Future studies should investigate the effects of TPE on specific pro-inflammatory immune cell subsets and on cytokines beyond IL-6 to understand the mode of action in neuroinflammatory disorders that are presumably not antibody-mediated.

Moreover, we have toned down the conclusions throughout the discussion because of the small number of patients included into this pilot study and emphasized more clearly the limitation of the small cohort. However, according to the findings from our pilot study, we cannot only make a statement about the extent and duration of antibody wash-out of TPE and also show that the here investigated soluble markers were surprisingly not eliminated, but according to this exploratory work we also encourage future studies to focus on particular soluble substances and on pro-inflammatory lymphocyte subsets.

Reviewer 2 Report

The author investigated the immunological changes induced by therapeutic plasma exchange in people affected by various neurological disorders. The study is of interest since it explores the mode of action of TPE giving clues to the understanding of the benefits determined by this technique and also about possible transient or long-term effects on the immune system.

The paper is well-written and the study design is appropriate. The main limit consists of the very small sample considered, even though this was expected due to the uncommon use of this procedure.

There are some issues that need to be solved:

The following sentences appear at the end of paragraph 2.4 in the Results section: "This section may be divided by subheadings. It should provide a concise and precise description of the experimental results, their interpretation, as well as the experimental conclusions that can be drawn." Do these sentences represent a mistake?

Figure 1 is presented with many coloured lines but no legend is provided to explain the different colours.

Figure 2 and 3: what does each line in every graph represents?

Discussion section, last sentence; there is a typo: "rational". 

Author Response

We would like to thank the reviewer for the time and the precious recommendations.

1. The following sentences appear at the end of paragraph 2.4 in the Results section: "This section may be divided by subheadings. It should provide a concise and precise description of the experimental results, their interpretation, as well as the experimental conclusions that can be drawn." Do these sentences represent a mistake?

Thank You. Yes, it was by mistake, and we have deleted the respective paragraph.

2. Figure 1 is presented with many coloured lines but no legend is provided to explain the different colours.

Thank You for this important point. We have now changed the colour scheme in figure 1, so that each colour represents a particular pathogen-specific antibody. We have now added a legend to explain the colours.

3. Figure 2 and 3: what does each line in every graph represents?

Figures 2 and 3 represent the raw data set, and each line represents and the corresponding symbols the data of one patient. We have added the following sentence to the respective legends for better clarification:

Symbols and connecting lines show the raw data of each patient included at the respective time of sampling.

4. Discussion section, last sentence; there is a typo: "rational". 

Thank you very much, we have corrected it.

Round 2

Reviewer 1 Report

In this study, the authors explored changes of soluble, humoral, and cellular immune components associated with TPE in ten patients with neurological autoimmune disorders, and pathogen-specific antibody levels were reduced post-TPE and recovered in the follow-ups. Mean IL-6 and CRP concentrations increased while PCT 24 remained unaffected. Furthermore, no alterations in B- and T-cell populations. No adverse events related to TPE occurred. Although the conclusions were just concluded from 10 patients, it can still provide some useful information for clinical. More samples and further investigations should be analyzed in future.

Reviewer 2 Report

the authors addressed properly all the comments made.